# Binarized Spectral Compressive Imaging

**Yuanhao Cai** [1]**, Yuxin Zheng** [1]**, Jing Lin** [1]**,**
**Xin Yuan** [2]**, Yulun Zhang** [3,*] **, Haoqian Wang** [1,*]
[1] Tsinghua University, [2] Westlake University, [3] ETH Zürich

## Abstract

Existing deep learning models for hyperspectral image (HSI) reconstruction achieve good performance but require powerful hardwares with enormous memory and computational resources. Consequently, these methods can hardly be deployed on resource-limited mobile devices. In this paper, we propose a novel method, Binarized Spectral-Redistribution Network (BiSRNet), for efficient and practical HSI restoration from compressed measurement in snapshot compressive imaging (SCI) systems. Firstly, we redesign a compact and easy-to-deploy base model to be binarized. Then we present the basic unit, Binarized Spectral-Redistribution Convolution (BiSR-Conv). BiSR-Conv can adaptively redistribute the HSI representations before binarizing activation and uses a scalable hyperbolic tangent function to closer approximate the Sign function in backpropagation. Based on our BiSR-Conv, we customize four binarized convolutional modules to address the dimension mismatch and propagate full-precision information throughout the whole network. Finally, our BiSRNet is derived by using the proposed techniques to binarize the base model. Comprehensive quantitative and qualitative experiments manifest that our proposed BiSRNet outperforms state-of-the-art binarization algorithms. Code and models are publicly available at https://github.com/caiyuanhao1998/BiSCI

## 1 Introduction

Compared to normal RGB images, hyperspectral images (HSIs) have more spectral bands to capture richer information of the desired scenes. Thus, HSIs have wide applications in agriculture [1, 2, 3], medical image analysis [4, 5, 6], object tracking [7, 8, 9], remote sensing [10, 11, 12], *etc.*

To capture HSIs, conventional imaging systems leverage 1D or 2D spectrometers to scan the desired scenes along the spatial or spectral dimension. Yet, this process is very time-consuming and thus fails in measuring dynamic scenes. In recent years, snapshot compressive imaging (SCI) systems [13, 14, 15, 16, 17] have been developed to capture HSI cubes in real time. Among these SCI systems, the coded aperture snapshot spectral imaging (CASSI) [14, 18, 19] demonstrates its outstanding effectiveness and efficiency. The CASSI systems firstly employ a coded aperture (physical mask) to modulate the 3D HSI cube, then use a disperser to shift spectral information of different wavelengths, and finally integrate these HSI signals on a detector array to capture a 2D compressed measurement. We study the inverse problem, *i.e.*, restoring the original 3D HSI cube from the 2D measurement.

Existing state-of-the-art (SOTA) SCI reconstruction methods are based on deep learning. Convolutional neural network (CNN) [18, 20, 21, 22, 23, 24, 25] and Transformer [26, 27, 28, 29] have been used to implicitly learn the mapping from compressed measurements to HSIs. Although superior performance is achieved, these CNN-/Transformer-based methods require powerful hardwares with abundant computing and memory resources, such as high-end graphics processing units (GPUs). However, edge devices (*e.g.,* mobile phones, hand-held cameras, small drones, *etc.*) evidently cannot meet the requirements of these expensive algorithms because edge devices have very limited memory, computational power, and battery. As mobile devices are more and more widely used, the demands of running and storing HSI restoration models on edge devices grow significantly. This motivates us to reduce the memory and computational burden of HSI reconstruction methods while preserving the performance as much as possible so that the algorithms can be deployed on resource-limited devices.

---

[*] Yulun Zhang and Haoqian Wang are the corresponding authors.

37th Conference on Neural Information Processing Systems (NeurIPS 2023).

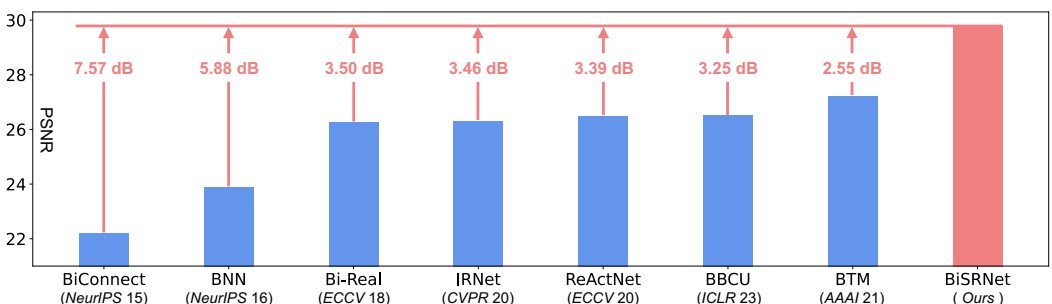

Figure 1: Comparison between our BiSRNet (in red color) and state-of-the-art BNNs (in blue color). BiSRNet significantly advances BTM [30], BBCU [31], ReActNet [32], IRNet [33], Bi-Real [34], BNN [35], and BiConnect [36] by 2.55, 3.25, 3.39, 3.46, 3.50, 5.88, and 7.57 dB on the simulation HSI reconstruction task.

The studies on neural network compression and acceleration [37, 38] can be divided into four categories: quantization [31, 32, 33, 34, 35, 39], pruning [40, 41, 42], knowledge distillation [43, 44], and compact network design [45, 46, 47, 48]. Among these methods, binarized neural network (BNN) belonging to quantization stands out because it can extremely compress the memory and computational costs by quantizing the weights and activations of CNN to only 1 bit. In particular, BNN could achieve $32\times$ memory compression ratio and up to $58\times$ practical computational reduction on central processing units (CPUs) [49]. In addition, the pure logical computation (*i.e.*, XNOR and bit-count operations) of BNN is highly energy-efficient for embedded devices [50, 51]. However, directly applying model binarization for HSI reconstruction algorithms may encounter three issues. **(i)** The HSI representations have different density and distribution in different spectral bands. Equally binarizing the activations of different spectral channels may lead to the collapse of HSI features. **(ii)** Previous model binarization methods mainly adopt a piecewise linear [33, 35, 36] or quadratic [31, 32, 34] function to approximate the non-differentiable Sign function. Nonetheless, there still remain large approximation errors between them and Sign. **(iii)** How to tackle the dimension mismatch problem during feature reshaping while allowing full-precision information propagation in BNN has not been fully explored. Previous model binarization methods [32, 33, 34, 35, 36] mainly consider the feature downsampling situation in a backbone network for high-level vision tasks.

Bearing the above considerations in mind, we propose a novel BNN-based method, namely Binarized Spectral-Redistribution Network (BiSRNet) for efficient and practical HSI reconstruction. **Firstly**, we redesign a compact and easy-to-deploy base model to be binarized. Different from previous CNN-/Transformer-based methods, this base model does not include complex computations like unfolding inference and non-local self-attention that are difficult to implement on edge devices. Instead, our base model only uses convolutional units that can be easily replaced by XNOR and bit-count logical operations on resource-limited devices. **Secondly**, we develop the basic unit, Binarized Spectral-Redistribution Convolution (BiSR-Conv), used in model binarization. Specifically, BiSR-Conv can adapt the density and distribution of HSI representations in spectral dimension before binarizing the activation. Besides, BiSR-Conv employs a scalable hyperbolic tangent function to closer approximate the non-differentiable Sign function by arbitrarily reducing the approximation error. **Thirdly**, as the full-precision information is very critical in BNN and the input HSI is the only full-precision source, we use BiSR-Conv to build up four binarized convolutional modules that can handle the dimension mismatch issue during feature reshaping and simultaneously propagate full-precision information through all layers. **Finally**, we derive our BiSRNet by using the proposed techniques to binarize the base model. As shown in Fig. 1, BiSRNet outperforms SOTA BNNs by large margins, **over 2.5 dB**.

In a nutshell, our contributions can be summarized as follows:

**(i)** We propose a novel BNN-based algorithm BiSRNet for HSI reconstruction. To the best of our knowledge, this is the first work to study the binarized spectral compressive imaging problem.

**(ii)** We customize a new binarized convolution unit BiSR-Conv that can adapt the density and distribution of HSI representations and approximate the Sign function better in backpropagation.

**(iii)** We design four binarized convolutional modules to address the dimension mismatch issue during feature reshaping and propagate full-precision information through all convolutional layers.

**(iv)** Our BiSRNet dramatically surpasses SOTA BNNs and even achieves comparable performance with full-precision CNNs while requiring extremely lower memory and computational costs.

## 2 Related Work

### 2.1 Hyperspectral Image Reconstruction

Traditional HSI reconstruction methods [14, 52, 53, 54, 55, 56, 57, 58, 59, 60, 61] are mainly based on hand-crafted image priors. Yet, these traditional methods achieve unsatisfactory performance and generality due to their poor representing capacity. Recently, deep CNN [18, 20, 21, 22, 23, 24, 25, 62] and Transformer [26, 27, 28, 29] have been employed as powerful models to learn the underlying mapping from compressed measurements to HSI data cubes. For example, TSA-Net [18] employs three spatial-spectral self-attention layers at the decoder of a U-shaped CNN. Cai *et al.* propose a series of Transformer-based algorithms (MST [26], MST++ [27], CST [28], and DAUHST [29]), pushing the performance boundary from 32 dB to 38 dB. Although impressive results are achieved, these CNN-/Transformer-based methods rely on powerful hardwares with enormous computational and memory resources, which are unaffordable for mobile devices. How to develop HSI restoration algorithms toward resource-limited platforms is under-explored. Our goal is to fill this research gap.

### 2.2 Binarized Neural Network

BNN [35] is the extreme case of model quantization as it quantizes the weights and activations into only 1 bit. Due to its impressive effectiveness in memory and computation compression, BNN has been widely applied in high-level vision [32, 33, 34, 35, 49] and low-level vision [30, 31, 39]. For example, Jiang *et al.* [30] train a BNN without batch normalization for image super-resolution. Xia *et al.* [31] design a binarized convolution unit BBCU for image super-resolution, denoising, and JPEG compression artifact reduction. Yet, the potential of BNN for SCI reconstruction has not been studied.

## 3 Method

### 3.1 Base Model

The full-precision model to be binarized should be compact and its computation should be easy to deploy on edge devices. However, previous CNN-/Transformer-based algorithms are computationally expensive or have large model sizes. Some of them exploit complex operations like unfolding inference [21, 22, 23, 29, 63] and non-local self-attention computation [18, 20, 26, 27, 28] that are challenging to binarize and difficult to implement on mobile devices. Hence, we redesign a simple, compact, and easy-to-deploy base model without using complex computation operations.

Inspired by the success of MST [26] and CST [28], we adopt a U-shaped structure for the base model as shown in Fig. 2. It consists of an encoder $\mathcal{E}$, a bottleneck $\mathcal{B}$, and a decoder $\mathcal{D}$. Please refer to the supplementary for the CASSI mathematical model. Firstly, we reverse the dispersion of CASSI by shifting back the measurement $\mathbf{Y} \in \mathbb{R}^{H \times (W + d(N_\lambda - 1)) \times N_\lambda}$ to derive the input $\mathbf{H} \in \mathbb{R}^{H \times W \times N_\lambda}$ as

$$\mathbf{H}(x, y, n_\lambda) = \mathbf{Y}(x, y - d(\lambda_n - \lambda_c)), \qquad (1)$$

where $H$, $W$, and $N_\lambda$ denote the HSI's height, width, and number of wavelengths. $d$ represents the shifting step. The concatenation of $\mathbf{H}$ and the 3D mask $\mathbf{M} \in \mathbb{R}^{H \times W \times N_\lambda}$ is fed into a feature embedding module to produce the shallow feature $\mathbf{X}_s \in \mathbb{R}^{H \times W \times N_\lambda}$. The feature embedding module is a $conv1 \times 1$ (convolutional layer with kernel size = $1 \times 1$). Subsequently, $\mathbf{X}_s$ undergoes the encoder $\mathcal{E}$, bottleneck $\mathcal{B}$, and decoder $\mathcal{D}$ to generate the deep feature $\mathbf{X}_d \in \mathbb{R}^{H \times W \times N_\lambda}$. $\mathcal{E}$ consists of two convolutional blocks and two downsample modules. The details of the convolutional block are depicted in Fig. 2 (b). The fusion up and down modules are both $conv1 \times 1$ to aggregate the feature maps and modify the channels. The downsample module is a strided $conv4 \times 4$ layer that downscales the feature maps and doubles the channels. $\mathcal{B}$ is a convolutional block. $\mathcal{D}$ consists of two convolutional blocks and two upsample modules. The upsample module is a bilinear interpolation followed by a $conv3 \times 3$ to upscale the feature maps and halve the channels. Skip connections between $\mathcal{E}$ and $\mathcal{D}$ are employed to alleviate the information loss during rescaling. Finally, the sum of $\mathbf{X}_s$ and $\mathbf{X}_d$ is fed into the feature mapping module ($conv1 \times 1$) to produce the reconstructed HSI $\mathbf{H}' \in \mathbb{R}^{H \times W \times N_\lambda}$.

### 3.2 Binarized Spectral-Redistribution Convolution

The details of BiSR-Conv are illustrated in Fig. 2 (c). We define the input full-precision activation as $\mathbf{X}_f \in \mathbb{R}^{H \times W \times C}$. We notice that HSI signals have different density and distribution along the spectral dimension due to the constraints of specific wavelengths. To adaptively fit this HSI nature, we propose to redistribute the HSI representations in channel wise before binarizing the activation as

$$\mathbf{X}_r = \boldsymbol{k} \cdot \mathbf{X}_f + \boldsymbol{b}, \qquad (2)$$

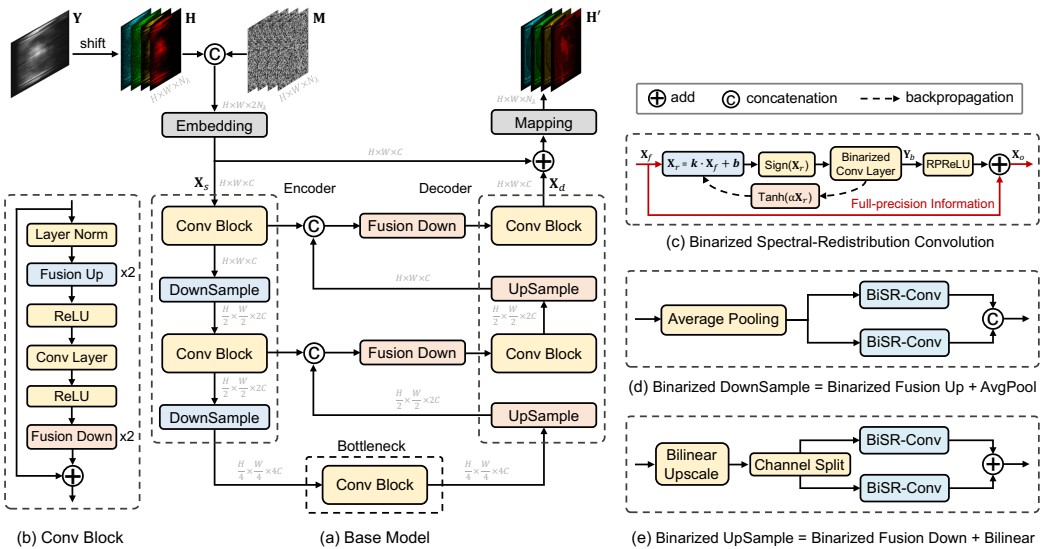

Figure 2: The overall diagram of our method. (a) The proposed base model to be binarized adopts a U-shaped architecture. (b) The components of the convolutional block. (c) The details of our Binarized Spectral-Redistribution Convolution (BiSR-Conv). (d) The structure of our binarized downsample module. The binarized fusion up module is derived by removing the average pooling operation. (e) The architecture of our binarized upsample module, which includes one more bilinear upscaling operation than the binarized fusion down module.

where $\mathbf{X}_r \in \mathbb{R}^{H \times W \times C}$ denotes the redistributed activation of $\mathbf{X}_f$. $\boldsymbol{k}$ and $\boldsymbol{b} \in \mathbb{R}^C$ are learnable parameters. $\boldsymbol{k}$ rescales the density of HSIs while $\boldsymbol{b}$ shifts the bias. Then $\mathbf{X}_r$ undergoes a Sign function to be binarized into 1-bit activation $\mathbf{X}_b \in \mathbb{R}^{H \times W \times C}$, where $x_b = +1$ or $-1$ for $\forall\, x_b \in \mathbf{X}_b$ as

$$x_b = \text{Sign}(x_r) = \begin{cases} +1, & x_r > 0 \\ -1, & x_r \leq 0 \end{cases} \tag{3}$$

where $x_r \in \mathbf{X}_r$. As shown in Fig. 3 (b) and (c), since the Sign function is non-differentiable, previous methods either adopt a piecewise linear function Clip($x$) [33, 35, 36, 49, 64] or a piecewise quadratic function Quad($x$) [31, 32, 34] to approximate the Sign function during the backpropagation as

$$\text{Clip}(x) = \begin{cases} +1, & x \geq 1 \\ x, & -1 < x < 1 \\ -1, & x \leq -1 \end{cases} \qquad \text{Quad}(x) = \begin{cases} +1, & x \geq 1 \\ 2x + x^2, & 0 < x < 1 \\ 2x - x^2, & -1 < x \leq 0 \\ -1, & x \leq -1 \end{cases} \tag{4}$$

Nonetheless, the Clip function is a rough estimation and there is a large approximation error between Clip and Sign. The shaded areas in Fig. 3 reflect the differences between the Sign function and its approximations. The shaded area corresponding to the Clip function is 1. Besides, once the absolute values of weights or activations are outside the range of $[-1, 1]$, they are no longer updated. Although the piecewise quadratic function is a closer approximation (the shaded area is 2/3) than Clip, the above two problems have not been fundamentally resolved. To address the two issues, we redesign a scalable hyperbolic tangent function to approximate the Sign function in the backpropagation as

$$x_b = \text{Tanh}(\alpha x_r) = \frac{e^{\alpha x_r} - e^{-\alpha x_r}}{e^{\alpha x_r} + e^{-\alpha x_r}}, \tag{5}$$

where $\alpha \in \mathbb{R}^+$ is a learnable parameter adaptively adjusting the distance between Tanh($\alpha x$) and Sign($x$). $e$ denotes the natural constant. We prove that when $\alpha \to +\infty$, Tanh($\alpha x$) $\to$ Sign($x$) as

$$\lim_{\alpha \to +\infty} \text{Tanh}(\alpha x) = \begin{cases} \lim\limits_{\alpha \to +\infty} \dfrac{e^{\alpha x} - 0}{e^{\alpha x} + 0} & = +1, \quad x > 0 \\[2mm] \lim\limits_{\alpha \to +\infty} \dfrac{e^0 - e^0}{e^0 + e^0} & = \;\;0, \quad x = 0 \\[2mm] \lim\limits_{\alpha \to +\infty} \dfrac{0 - e^{-\alpha x}}{0 + e^{-\alpha x}} & = -1, \quad x < 0 \end{cases} \tag{6}$$

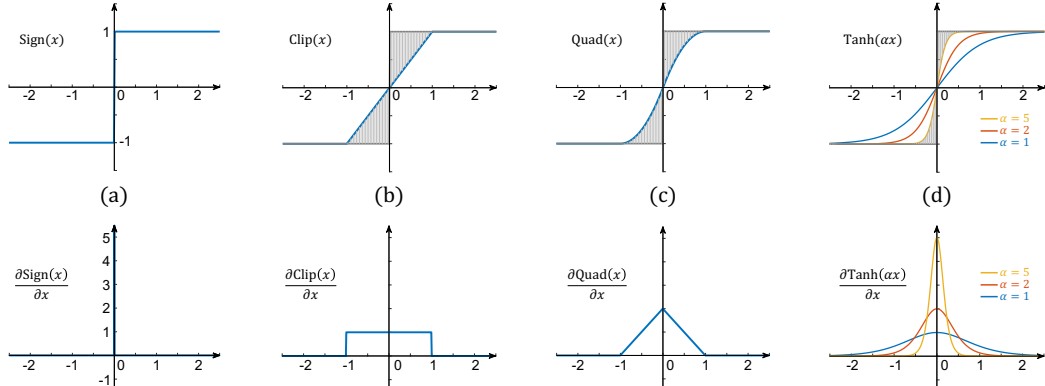

Figure 3: The upper line shows (a) Sign($x$) and its three approximation functions including (b) piecewise linear function Clip($x$), (c) piecewise quadratic function Quad($x$), and (d) our scalable hyperbolic tangent function Tanh($\alpha x$). The area of the shaded region reflects the approximation error. The lower line depicts the derivatives.

If strictly following the mathematical definition, Sign(0) = 0 $\neq$ $\pm$1. However, in BNN, the weights and activations are binarized into 1-bit, *i.e.*, only two values ($\pm$1). Hence, Sign(0) is usually set to $-1$. Similar to this common setting, we also define $\lim\limits_{\alpha \to +\infty}$ Tanh($\alpha \cdot 0$) = $-1$ in BNN. Then we have

$$\lim_{\alpha \to +\infty} \text{Tanh}(\alpha x) = \text{Sign}(x). \tag{7}$$

We compute the area of the shaded region between our Tanh($\alpha x$) and Sign($x$) in Fig. 3 (d) as

$$\int_{-\infty}^{+\infty} |\text{Sign}(x) - \text{Tanh}(\alpha x)| \, \mathrm{d}x = 2 \int_{0}^{+\infty} (1 - \text{Tanh}(\alpha x)) \, \mathrm{d}x$$

$$= 2(x - x + \frac{1}{\alpha}\log(\text{Tanh}(\alpha x) + 1))\big|_{x=0}^{x=+\infty} \tag{8}$$

$$= \frac{2}{\alpha}(\log(2) - \log(1)) = \frac{2\log(2)}{\alpha}.$$

Different from previous Clip($x$) and Quad($x$), our Tanh($\alpha x$) can arbitrarily reduce the approximation error with Sign($x$) when $\alpha$ in Eq. (8) is large enough. Besides, our Tanh($\alpha x$) is neither piecewise nor unchanged when $x$ is outside the range of $[-1, 1]$. On the contrary, the weights and activations can still be updated when their absolute values are larger than 1. In addition, as depicted in the lower line of Fig. 3, the value ranges $[0, 1]$ and fixed shapes of $\frac{\partial \text{Clip}(x)}{\partial x}$ and $\frac{\partial \text{Quad}(x)}{\partial x}$ are fundamentally different from those of $\frac{\partial \text{Sign}(x)}{\partial x} \in [0, +\infty)$. In contrast, our $\frac{\partial \text{Tanh}(\alpha x)}{\partial x}$ can change its value range $(0, \alpha)$ and shape by adapting the parameter $\alpha$. It is more flexible and can approximate $\frac{\partial \text{Sign}(x)}{\partial x}$ better.

In the binarized convolutional layer, the 32-bit weight $\mathbf{W}_f$ is also binarized into 1-bit weight $\mathbf{W}_b$ as

$$w_b = \mathbb{E}_{w_f \in \mathbf{W}_f}(|w_f|) \cdot \text{Sign}(w_f), \tag{9}$$

where $\mathbb{E}$ represents computing the mean value. Multiplying the mean absolute value of 32-bit weight value $w_f \in \mathbf{W}_f$ can narrow down the difference between binarized and full-precision weights. Subsequently, the computationally heavy operations of floating-point matrix multiplication in full-precision convolution can be replaced by pure logical XNOR and bit-count operations [49] as

$$\mathbf{Y}_b = \mathbf{X}_b * \mathbf{W}_b = \text{bit-count}(\text{XNOR}(\mathbf{X}_b, \mathbf{W}_b)), \tag{10}$$

where $\mathbf{Y}_b$ represents the output and $*$ denotes the convolution operation. Since the value range of full-precision activation $\mathbf{X}_f$ largely varies from that of 1-bit convolution output $\mathbf{Y}_b$, directly employing an identity mapping to aggregate them may cover up the information of $\mathbf{Y}_b$. To cope with this problem, we first fed $\mathbf{Y}_b$ into a RPReLU [32] activation function to change its value range and then add it with $\mathbf{X}_f$ by a residual connection to propagate full-precision information as

$$\mathbf{X}_o = \mathbf{X}_f + \text{RPReLU}(\mathbf{Y}_b), \tag{11}$$

where $\mathbf{X}_o$ denotes the output feature and RPReLU is formulated for the $i$-th channel of $\mathbf{Y}_b$ as

$$\text{RPReLU}(y_i) = \begin{cases} y_i - \gamma_i + \zeta_i, & y_i > \gamma_i \\ \beta_i \cdot (y_i - \gamma_i) + \zeta_i, & y_i \le \gamma_i \end{cases} \tag{12}$$

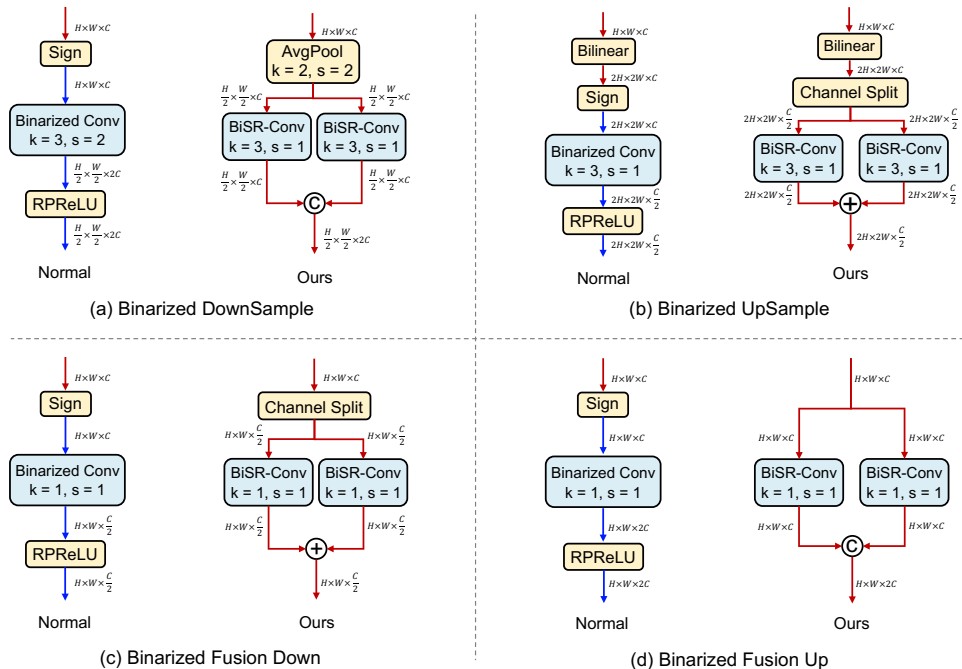

Figure 4: Comparison between normal and our binarized convolutional modules, including (a) downsample, (b) upsample, (c) fusion down to half the channels, and (d) fusion up to double the channels. The red arrow ↓ indicates the full-precision information flow, while the blue arrow ↓ denotes the binarized signal flow.

where $y_i \in \mathbb{R}$ indicates single pixel values belonging to the $i$-th channel of $\mathbf{Y}_b$. $\beta_i, \gamma_i$, and $\zeta_i \in \mathbb{R}$ represents learnable parameters. Please note that the full-precision information is not blocked by the binarized convolutional layer in the proposed BiSR-Conv. Instead, it is propagated by the bypass identity mapping, as shown in the red arrow → in Fig. 2 (c). Based on this important property of BiSR-Conv, we design the four binarized convolutional modules, as illustrated in Fig. 2 (d) and (e).

### 3.3 Binarized Convolutional Modules

In model binarization, the identity mapping is critical to propagate full-precision information and ease the training procedure. The only source of full-precision information is the input end ($\mathbf{X}_s$ in Fig. 2) of the binarized part. However, the dimension mismatch during the feature downsampling, upsampling, and aggregation processes blocks the residual connections, which degrades the HSI reconstruction performance. To tackle this problem, we use BiSR-Conv to build up four binarized convolutional modules including downsample, upsample, fusion up, and fusion down with unblocked identity mappings to make sure the full-precision information can flow through all binarized layers.

Specifically, Fig. 4 compares the normal and our binarized modules. The red arrow ↓ indicates the full-precision information flow, while the blue arrow ↓ denotes the binarized signal flow. The downsample modules in Fig. 4 (a) downscale the input feature maps and double the channels. The upsample modules in Fig. 4 (b) upscale the input spatial dimension and half the channels. The fusion down modules in Fig. 4 (c) maintain the spatial size of the input feature and half the channels. The fusion up modules in Fig. 4 (d) keep the spatial dimension of the input feature maps while doubling the channels. In the normal modules, the full-precision information is blocked by the Sign function and binarized into 1-bit signal. Meanwhile, the intermediate feature maps are directly reshaped by the binarized convolutional layers. For example, in the normal binarized downsample module, the intermediate feature is directly reshaped from $\mathbb{R}^{H \times W \times C}$ to $\mathbb{R}^{\frac{H}{2} \times \frac{W}{2} \times 2C}$ by a strided 1-bit $conv4 \times 4$. The spatial and channel dimension mismatch of the input and output feature maps impede the identity mapping to propagate full-precision information from previous layers. In contrast, our binarized modules rely on the proposed BiSR-Conv that has a bypass identity connection for full-precision information flow, as shown in Fig. 2 (c). By using channel-wise concatenating and splitting operations, the intermediate feature maps at the input and output ends of BiSR-Conv are free from being reshaped. Therefore, the full-precision information can flow through our binarized modules. Finally, we derive our BiSRNet by using BiSR-Conv and the four modules to binarize $\mathcal{E}, \mathcal{B}$, and $\mathcal{D}$ of the base model.

| Algorithms | Bit | Category | Params (K) | OPs (G) | S1 | S2 | S3 | S4 | S5 | S6 | S7 | S8 | S9 | S10 | Avg |
|---|---|---|---|---|---|---|---|---|---|---|---|---|---|---|---|
| TwIST [59] | 64 | Model | - | - | 25.16 0.700 | 23.02 0.604 | 21.40 0.711 | 30.19 0.851 | 21.41 0.635 | 20.95 0.644 | 22.20 0.643 | 21.82 0.650 | 22.42 0.690 | 22.67 0.569 | 23.12 0.669 |
| GAP-TV [56] | 64 | Model | - | - | 26.82 0.754 | 22.89 0.610 | 26.31 0.802 | 30.65 0.852 | 23.64 0.703 | 21.85 0.688 | 23.76 0.655 | 21.98 0.682 | 22.63 0.584 | 23.10 0.669 | 24.36 0.669 |
| DeSCI [53] | 64 | Model | - | - | 27.13 0.748 | 23.04 0.620 | 26.62 0.818 | 34.96 0.897 | 23.94 0.706 | 22.38 0.683 | 24.45 0.743 | 22.03 0.673 | 24.56 0.732 | 23.59 0.587 | 25.27 0.721 |
| $\lambda$-Net [20] | 32 | CNN | 62640 | 117.98 | 30.10 0.849 | 28.49 0.805 | 27.73 0.870 | 37.01 0.934 | 26.19 0.817 | 28.64 0.853 | 26.47 0.806 | 26.09 0.831 | 27.50 0.826 | 27.13 0.816 | 28.53 0.841 |
| TSA-Net [18] | 32 | CNN | 44250 | 110.06 | 32.03 0.892 | 31.00 0.858 | 32.25 0.915 | 39.19 0.953 | 29.39 0.884 | 31.44 0.908 | 30.32 0.878 | 29.35 0.888 | 30.01 0.890 | 29.59 0.874 | 31.46 0.894 |
| BiConnect [36] | 1 | BNN | 35 | 1.18 | 25.85 0.676 | 22.07 0.530 | 18.92 0.558 | 25.18 0.636 | 21.21 0.568 | 21.82 0.547 | 21.84 0.570 | 22.25 0.580 | 19.57 0.556 | 23.18 0.524 | 22.19 0.575 |
| BNN [35] | 1 | BNN | 35 | 1.18 | 26.69 0.661 | 23.98 0.551 | 20.58 0.566 | 28.53 0.679 | 22.96 0.584 | 24.12 0.599 | 23.20 0.568 | 23.29 0.590 | 21.65 0.588 | 23.86 0.547 | 23.88 0.593 |
| Bi-Real [34] | 1 | BNN | 35 | 1.18 | 28.06 0.701 | 26.05 0.644 | 24.92 0.654 | 31.04 0.733 | 25.32 0.664 | 26.54 0.671 | 25.09 0.631 | 25.47 0.678 | 24.69 0.644 | 25.41 0.622 | 26.26 0.664 |
| IRNet [33] | 1 | BNN | 35 | 1.18 | 27.91 0.700 | 25.84 0.620 | 25.27 0.661 | 31.77 0.723 | 25.12 0.663 | 26.31 0.685 | 25.29 0.665 | 25.14 0.662 | 25.07 0.668 | 25.20 0.603 | 26.30 0.665 |
| ReActNet [32] | 1 | BNN | 36 | 1.18 | 27.91 0.707 | 26.17 0.633 | 25.40 0.682 | 31.58 0.725 | 25.43 0.675 | 26.43 0.670 | 25.85 0.703 | 25.50 0.650 | 25.47 0.677 | 25.11 0.583 | 26.48 0.671 |
| BBCU [31] | 1 | BNN | 36 | 1.18 | 27.91 0.706 | 26.21 0.628 | 25.44 0.654 | 31.33 0.741 | 25.30 0.677 | 26.68 0.704 | 25.42 0.668 | 25.59 0.671 | 25.69 0.670 | 25.59 0.615 | 26.51 0.673 |
| BTM [30] | 1 | BNN | 36 | 1.18 | 28.75 0.739 | 26.91 0.674 | 26.14 0.708 | 32.74 0.794 | 25.87 0.692 | 27.37 0.739 | 26.26 0.707 | 26.20 0.718 | 26.10 0.717 | 25.73 0.671 | 27.21 0.716 |
| **BiSRNet (Ours)** | 1 | BNN | 36 | 1.18 | **30.95 0.847** | **29.21 0.791** | **29.11 0.828** | **35.91 0.903** | **28.19 0.827** | **30.22 0.863** | **27.85 0.800** | **28.82 0.843** | **29.46 0.832** | **27.88 0.800** | **29.76 0.837** |

Table 1: Quantitative results of BiSRNet, seven 1-bit BNN-based methods, two 32-bit CNN-based algorithms, and three 64-bit model-based methods on 10 scenes (S1~S10) of the KAIST [65] dataset. Params, OPs, PSNR (upper entry in each cell), and SSIM (lower entry in each cell) are reported.

## 4 Experiment

### 4.1 Experimental Settings

Following [18, 26, 28, 29, 63], we select 28 wavelengths from 450nm to 650nm by using spectral interpolation manipulation to derive HSIs. We conduct experiments on simulation and real datasets.

**Simulation Data.** Two simulation datasets, CAVE [66] and KAIST [65], are adopted. The CAVE dataset provides 32 HSIs with a spatial size of 512×512. The KAIST dataset includes 30 HSIs at a spatial size of 2704×3376. We use CAVE for training and select 10 scenes from KAIST for testing.

**Real Data.** We adopt the five real HSIs captured by the CASSI system developed in [18] for testing.

**Evaluation Metrics.** The peak signal-to-noise ratio (PSNR) and structure similarity (SSIM) are adopted as metrics to evaluate the HSI reconstruction performance. Similar to [31, 32, 33, 34, 35], the operations per second of BNN ($OPs^b$) is computed as $OPs^b = OPs^f / 64$ ($OPs^f$ = FLOPs) to measure the computational complexity and the parameters of BNN is calculated as $Params^b = Params^f / 32$, where the superscript $b$ and $f$ refer to the binarized and full-precision models. The total computational and memory costs of a model are computed as $OPs = OPs^b + OPs^f$ and $Params = Params^b + Params^f$.

**Implementation Details.** The proposed BiSRNet is implemented by PyTorch [67]. We use Adam [68] optimizer ($\beta_1 = 0.9$ and $\beta_2 = 0.999$) and Cosine Annealing [69] scheduler to train BiSRNet for 300 epochs on a single RTX 2080 GPU. Training samples are patches with spatial sizes of 256×256 and 96×96 randomly cropped from 28-channel 3D HSI data cubes for simulation and real experiments. The shifting step $d$ is 2. The batch size is 2. We set the basic channel $C = N_\lambda = 28$ to store HSI information. We use random flipping and rotation for data augmentation. The training loss function is the root mean square error (RMSE) between reconstructed and ground-truth HSIs.

### 4.2 Quantitative Results

We compare our BiSRNet with 12 SOTA algorithms, including seven 1-bit BNN-based methods (BiConnect [36], BNN [35], Bi-Real [34], IRNet [33], ReActNet [32], BTM [30], and BBCU [31]), two 32-bit full-precision CNN-based methods ($\lambda$-Net [20] and TSA-Net [18]), and three 64-bit double-precision model-based methods (TwIST [59], GAP-TV [56], and DeSCI [53]). For a fair comparison, we set the Params and OPs of BNN-based methods to the same values.

Directly applying the seven SOTA BNN-based methods to HSI reconstruction task achieves unsatisfactory performance, ranging from 22.19 dB to 27.21 dB. Our BiSRNet surpasses these methods by large margins. More specifically, BiSRNet significantly outperforms BTM, BBCU, ReActNet, IRNet,

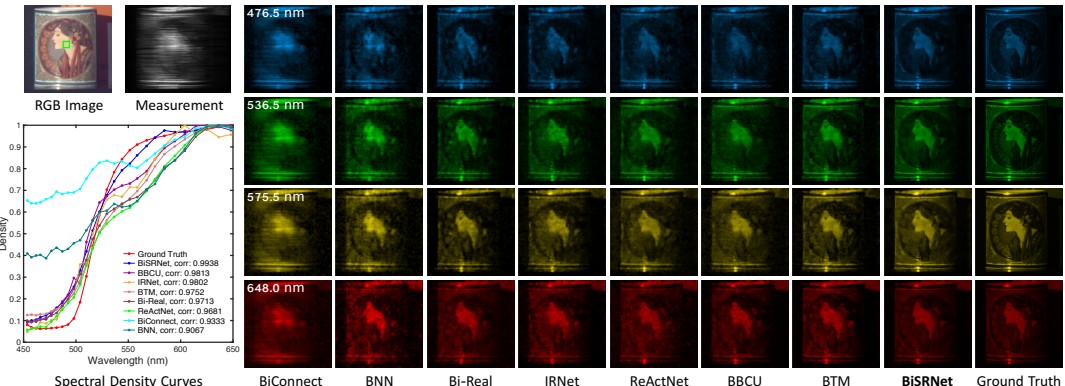

Figure 5: Reconstructed simulation HSIs of *Scene* 1 with 4 out of 28 spectral channels. Seven SOTA BNN-based algorithms and our proposed BiSRNet are compared. The spectral density curves (bottom-left) are corresponding to the selected region of the green box in the RGB image (Top-left). Please zoom in for a better view.

Bi-Real, BNN, and BiConnect by 2.55, 3.25, 3.39, 3.46, 3.50, 5.88, and 7.57 dB. This evidence suggests the significant effectiveness advantage of our BiSRNet in HSI restoration.

The proposed BiSRNet with extremely lower memory and computational complexity yields comparable results with 32-bit full-precision CNN-based methods. Surprisingly, our BiSRNet outperforms $\lambda$-Net by 1.23 dB while only costing 0.06 % (36/62640) Params and 1.0% (1.18/117.98) OPs. In contrast, the previous best BNN-based method BTM is still 1.33 dB lower than $\lambda$-Net. When compared with TSA-Net, our BiSRNet only uses 0.08% Params and 1.1% OPs but achieves 94.6% (29.76/31.46) performance. These results demonstrate the efficiency superiority of the proposed method.

The three model-based algorithms are implemented by MATLAB, where the default type of variable is double-precision floating-point number. Although they use more accurate 64-bit data type, our 1-bit BiSRNet dramatically outperforms DeSCI, GAP-TV, and TwIST by 4.06, 5.40, and 6.64 dB.

### 4.3 Qualitative Results

**Simulation HSI Restoration.** Fig. 5 depicts the simulation HSIs on *Scene* 1 with 4 out of 28 spectral channels reconstructed by the 7 SOTA BNN-based algorithms and BiSRNet. Previous BNNs are less favorable to restore HSI details. They generate blurry HSIs while introducing undesirable artifacts. In contrast, BiSRNet reconstructs more visually pleasing HSIs with more structural contents and sharper edges. Additionally, we plot the spectral density curves (bottom-left) corresponding to the selected regions of the green box in the RGB image (Top-left). BiSRNet achieves the highest correlation score with the ground truth, suggesting the advantage of BiSRNet in spectral-wise consistency restoration.

**Real HSI Restoration.** Fig. 6 visualizes the reconstructed HSIs of the seven SOTA BNN-based algorithms and our BiSRNet. We follow the setting of [18, 26, 28, 29, 63] to re-train the models with all samples of the CAVE and KAIST datasets. To simulate the noise disturbance in real imaging scenes, we inject 11-bit shot noise into measurements during training. It can be observed that our BiSRNet is more effective in detailed content reconstruction and real imaging noise suppression.

### 4.4 Ablation Study

**Break-down Ablation.** We adopt baseline-1 to conduct a break-down ablation towards higher performance. Baseline-1 is derived by using vanilla 1-bit convolution to replace BiSR-Conv and normal binarized convolutional modules (see Fig. 4) to replace our binarized convolutional modules. As shown in Tab. 2a, baseline-1 yields 23.90 dB in PSNR and 0.594 in SSIM. When we apply BiSR-Conv, the model achieves 3.90 dB improvement. Then we successively use our binarized downsample (BiDS), upsample (BiUS), fusion down (BiFD), and fusion up (BiFU) modules, the model gains by 1.96 dB in total. These results verify the effectiveness of the proposed techniques.

**BiSR-Conv.** To study the effects of BiSR-Conv components, we adopt baseline-2 to conduct an ablation. Baseline-2 is obtained by removing Spectral-Redistribution (SR) operation and Sign approximation Tanh($\alpha x$) from BiSRNet. As reported in Tab. 2b, baseline-2 achieves 27.68 dB in PSNR and 0.723 in SSIM. When we respectively apply SR and Tanh($\alpha x$), baseline-2 gains by 1.29 and 1.06 dB. When we exploit SR and Tanh($\alpha x$) jointly, the model achieves 2.08 dB improvement.

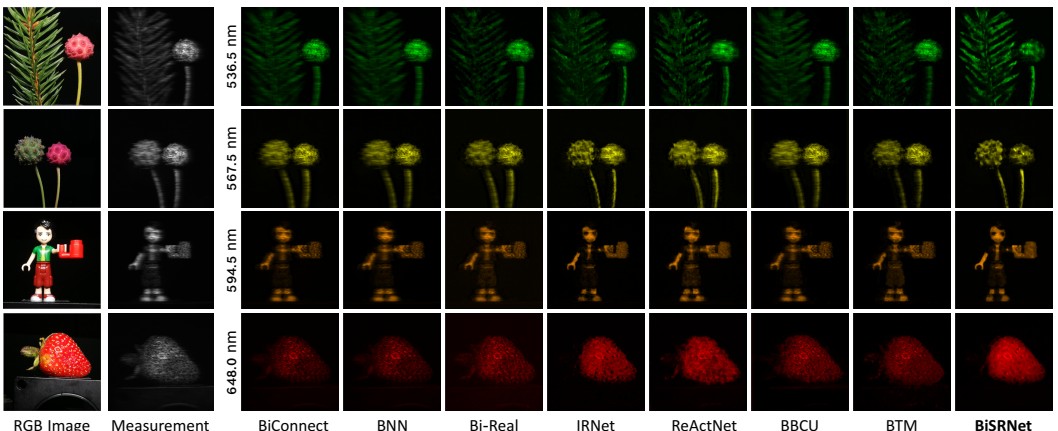

Figure 6: Reconstructed real HSIs of seven SOTA BNN-based algorithms and our BiSRNet on four scenes with 4 out of 28 wavelengths. BiSRNet is more effective in reconstructing detailed contents and suppressing noise.

| Method | Baseline-1 | +BiSR-Conv | +BiDS | +BiUS | +BiFD | +BiFU |
|--------|-----------|-----------|-------|-------|-------|-------|
| PSNR | 23.90 | 27.80 | 27.97 | 28.07 | 28.31 | **29.76** |
| SSIM | 0.594 | 0.737 | 0.729 | 0.758 | 0.776 | **0.837** |
| OPs (M) | 1176 | 1176 | 1176 | 1176 | 1176 | 1176 |
| Params (K) | 34.82 | 35.48 | 35.49 | 35.51 | 35.72 | 35.81 |

(a) Break-down ablation study towards higher performance

| Baseline-2 | SR | Tanh($\alpha x$) | PSNR | SSIM |
|-----------|----|-----------------|------|------|
| ✓ | | | 27.68 | 0.723 |
| ✓ | ✓ | | 28.97 | 0.783 |
| ✓ | | ✓ | 28.74 | 0.782 |
| ✓ | ✓ | ✓ | **29.76** | **0.837** |

(b) Ablation study of BiSR-Conv

| Method | PSNR | SSIM |
|--------|------|------|
| Clip($x$) | 28.97 | 0.783 |
| Quad($x$) | 29.02 | 0.794 |
| Tanh($\alpha x$) | **29.76** | **0.837** |

(c) Study of approximation

| Binarized Part | OPs$^f$ (M) | OPs$^b$ (M) | Params$^f$ | Params$^b$ | PSNR$^b$ | SSIM$^b$ |
|----------------|-----------|-----------|-----------|-----------|---------|---------|
| Encoder $\mathcal{E}$ | 3390 | 53 | 177878 | 5559 | 32.28 | 0.905 |
| Bottleneck $\mathcal{B}$ | 1096 | 17 | 278889 | 8715 | 33.80 | 0.932 |
| Decoder $\mathcal{D}$ | 5005 | 78 | 186562 | 5830 | 33.03 | 0.919 |

(d) Ablation study of binarizing different parts of the base model

Table 2: Ablations on the simulation datasets. In table (a), BiUS, BiDS, BiFU, and BiFD denote the binarized upsample, downsample, fusion up, and fusion down modules of Fig. 4. In table (b), SR refers to the Spectral-Redistribution of Eq. (11). In table (d), the full-precision model yields 34.11 dB in PSNR and 0.936 in SSIM.

**Sign Approximation.** We compare our scalable hyperbolic tangent function with previous Sign approximation functions. The experimental results are listed in Tab. 2c. Our Tanh($\alpha x$) dramatically surpasses the piecewise linear function Clip($x$) and quadratic function Quad($x$) by 0.79 and 0.74 dB, suggesting the superiority of the proposed Tanh($\alpha x$). This advantage can be explained by the analysis in Sec. 3.2 that our Tanh($\alpha x$) is more flexible and can adaptively reduce the difference with Sign($x$).

**Binarizing Different Parts.** We binarize one part of the base model while keeping the other parts full-precision to study the binarization benefit. The experimental results are reported in Tab. 2d. The base model yields 34.11 dB in PSNR and 0.936 in SSIM while costing 10.52 G OPs and 634 K Params. It can be observed from Tab. 2d: **(i)** Binarizing the bottleneck $\mathcal{B}$ reduces the Params the most (270174) with the smallest performance drop (only 0.31 dB). **(ii)** Binarizing the decoder $\mathcal{D}$ achieves the largest OPs reduction (4927 M) while the performance degrades by a moderate margin (1.08 dB).

## 5 Conclusion

In this paper, we propose a novel BNN-based method BiSRNet for binarized HSI restoration. To the best of our knowledge, this is the first work to study the binarized spectral compressive imaging reconstruction problem. We first redesign a compact and easy-to-deploy base model with simple computation operations. Then we customize the basic unit BiSR-Conv for model binarization. BiSR-Conv can adaptively adjust the density and distribution of HSI representations before binarizing the activation. Besides, BiSR-Conv employs a scalable hyperbolic tangent function to closer approach Sign by arbitrarily reducing the approximation error. Subsequently, we use BiSR-Conv to build up four binarized convolutional modules that can handle the dimension mismatch issue during feature reshaping and propagate full-precision information through all layers. Comprehensive quantitative and qualitative experiments demonstrate that our BiSRNet significantly outperforms SOTA BNNs and even achieves comparable performance with full-precision CNN-based HSI reconstruction algorithms.

## Acknowledgement

This research was funded through National Key Research and Development Program of China (Project No. 2022YFB36066), in part by the Shenzhen Science and Technology Project under Grant (JCYJ20220818101001004, JSGG20210802153150005), National Natural Science Foundation of China (62271414), Science Fund for Distinguished Young Scholars of Zhejiang Province (LR23F010001), and Research Center for Industries of the Future at Westlake University.

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
