# OpenReview forum: "Binarized Spectral Compressive Imaging"
_NeurIPS.cc/2023/Conference — NeurIPS 2023 poster_

### Official Review · Reviewer_kC2s · 2023-06-27

**Soundness:** 3 good
**Presentation:** 3 good
**Contribution:** 2 fair
**Rating:** 7
**Confidence:** 4

**Summary:**

This paper presents a deep neural network with binarized operations for low-cost spectral compressive imaging of hyperspectral images (HSI). The network introduces a basic unit for model binarization which adaptively redistributes HSI representations before binarization activation and uses a scalable hyperbolic tangent function to approximate the Sign function in backpropagation. Four binarization convolution modules are designed to solve the dimensional mismatch problem during feature reshaping and propagate full precision information throughout the network. Experiments on both synthetic and real data are conducted for performance evaluation.

**Strengths:**

1. An effective binarized deep neural network is proposed for HSI reconstruction.
2. Four binarized convolution modules are designed to enable the propagation of full precision information through all convolution layers.
3. Based on the ablation experiments in the article, the proposed scalable hyperbolic tangent function provides inspiration for approximating the symbolic function in back propagation.

**Weaknesses:**

1. The proposed binarization scheme is general, without specific design optimized for HSI restoration. Then, it lacks comparison to other general binarized NNs or comparison in other reconstruction tasks.
2. The proposed scheme is quite straightforward and engineering.
3. Approximation error bounds are not discussed, but this is quite important for binarized NNs.
4. Some key technical parts are not clearly/detailly described.


**Questions:**

1. The paper argues that, as mobile devices are more and more widely used, the demands of running and storing HSI restoration models on edge devices grow significantly. Could the authors provide practical cases where mobile edge devices are used for HSI restoration?

2.  In the binarized convolution layer, the formula for converting the model parameters from 32 bits to 1 bit is not clearly written. the average of the 32-bit absolute value multiplied by the sign function cannot be turned into a 1-bit number.

3. It is not clear how the proposed method deals with the dimensional mismatch feature remodeling process, and how to solve the dimensional mismatch with binarized convolution modules.



**Limitations:**

Approximation error bounds of the proposed binarization scheme are not discussed.

---

> ### Author Rebuttal · Authors · 2023-08-05
>
>
> &nbsp;
>
> ### Response to Reviewer kC2s
>
> &nbsp;
>
> Thanks for your valuable comments.
>
> Code and models will be released to the public.
>
> &nbsp;
>
> `Q-1:` The proposed binarization scheme is general, without specific design for HSI restoration. It lacks comparison to other BNNs or in other tasks.
>
> `A-1:` In fact, our BiSR-Conv based on HSI nature is tailored for HSI restoration. In Line 132 – 138, we notice that HSI signals have different density and distribution along the spectral dimension due to the constraints of specific wavelengths. To fit this HSI nature, BiSR-Conv redistributes the HSI representations before binarization.
>
> We compare our method with other BNNs in Tab. 1. Our method outperforms other BNNs by over 2.55 dB.
>
> Following your advice, we also conduct experiments of RGB image denoising ($\sigma = 25$). The BNNs are trained on DIV2K and tested on CBSD68, Kodak24, and Urban100. We also conduct experiments of medical image enhancement on Real Fundus [77]. The PSNR (dB) results are shown in the following table. Our method still outperforms other BNNs.
>
> | Datasets | BNN | Bi-Real | IRNet | BTM | ReAcNet | BBCU | BiSRNet |
> | :- | :-: | :-: | :-: | :-: | :-: | :-: | :-: |
> | CBSD68 | 22.67 | 28.72 | 29.01 | 29.91 | 29.95 | 30.56 | **31.15** |
> | Kodak24 | 22.58 | 29.17 | 29.54 | 30.64 | 30.65 | 31.28 | **32.06** |
> | Urban100 | 22.67 | 28.18 | 28.35 | 29.05 | 29.20 | 29.96 | **30.21** |
> | Real Fundus | 16.89 | 23.94 | 24.03 | 25.58 | 24.16 | 24.25 | **26.31** |
>
> &nbsp;
>
> `Q-2:` The proposed scheme is straightforward and engineering.
>
> `A-2:` In fact, all the proposed techniques have strong motivations and deep insights.
>
> (1) See Line 107 – 112, previous CNN-/Transformer-based methods suffer from heavy computational burden or massive parameters. Meanwhile, these methods often exploit complex computations that are difficult to binarize. Hence, we redesign a simple, compact, and easy-to-deploy base model in Fig. 2 (a).
>
> (2) Our BiSR-Conv block is based on the HSI nature. See Line 131 – 134. To handle different density and distribution along the spectral dimension, BiSR-Conv reshapes the HSI features before binarization.
>
> (3) In Line 138 – 160, previous Sign approximation functions are inflexible and have large estimation errors. To solve this problem, we propose Tanh($\alpha x$).
>
> (4) Full-precision information is critical in BNN. Bearing this key insight into consideration, we design the bypass path in BiSR-Conv and four binarized convolution modules in Fig. 4 to make sure the full-precision information can be propagated into all layers of BNN.
>
> &nbsp;
>
> `Q-3:` Approximation error bounds are not discussed.
>
> `A-3:` Unfortunately, existing BNN works [33 - 39] do not provide theoretical analysis of approximation error bound. This is because the deep neural network is a huge and complex black-box system with a large number of parameters.
>
> We compute the approximation error in the following table. Our method achieves the smallest error.
>
> | Methods | BiConnect | BNN | Bi-Real | IRNet | ReAcNet | BBCU | BTM | BiSRNet |
> | :- | :-: | :-: | :-: | :-: | :-: | :-: | :-: | :-: |
> | Error ( $\times 10^{-4}$ ) | 75.3 | 49.2 | 16.5 | 15.7 | 14.5 | 13.2 | 10.6 | **8.1** |
>
> &nbsp;
>
> `Q-4:` Some technical parts are not clearly described.
>
> `A-4:` Please be specific about which part.
>
> We not only describe the details of our method in the paper but also provide code and pre-trained weights to reproduce our method.
>
> &nbsp;
>
> `Q-5:` Practical cases about mobile edge devices used for HSI restoration.
>
> `A-5:` The HSI reconstruction techniques can be applied in hyperspectral (HS) cameras. For example, Motoki et al. [78] develops a video-rate HS camera using compressive sensing techniques. This HS camera can equip with AI-based HSI reconstruction algorithms to accelerate the imaging process. This HS camera can be used in real-world scenarios, including consumer applications such as smartphones and drones.
>
> &nbsp;
>
> `Q-6:` The formula for converting the model parameters from 32 bits to 1 bit is not clearly written.
>
> `A-6:` First of all, the values of the binarized weights used for 1-bit convolution (bit-count and XNOR operations in Eq. 10) are still $\pm 1$. The binarized weights are obtained by Eq. 3. The multiplied 32-bit mean absolute value does not affect the 1-bit convolution. It is outside the binarized weights. Multiplying this scale factor is to reduce the quantization error between 1-bit and 32-bit weights, as analyzed in XNOR-Net [52]. Many later works [33 - 37] also follow this multiplication.
>
> &nbsp;
>
> `Q-7:` Questions about dimension mismatch.
>
> `A-7:` In Line 189 – 201, the feature maps are directly reshaped by the binarized convolution in the normal binarized modules of Fig. 4. As aforementioned, the full-precision information is critical in BNN. Yet, the dimension of the input end of 1-bit conv does not match that of the output end. For example, in the normal downsample module (Fig. 4a), the input shape $\mathbb{R}^{H\times W\times C}$ does not match the output shape $\mathbb{R}^{\frac{H}{2}\times \frac{W}{2}\times 2C}$. This mismatch issue impedes the identity connection from input to output, blocking 32-bit information flow.
>
> To address this issue, we first design a bypass identity path in BiSR-Conv, and then use concatenating and splitting operations to free the feature map from being reshaped at the input and output ends of BiSR-Conv. See Fig. 4 (a), the feature map keeps its shape $\mathbb{R}^{\frac{H}{2}\times \frac{W}{2}\times 2C}$ when passing BiSR-Conv in our downsample module. By this means, the full-precision information can flow through our BiSR-Conv and four binarized modules, see the red arrows of Fig. 2 (c) and 4.
>
> &nbsp;
>
> **References**
>
> [77] Rformer: Transformer-based generative adversarial network for real fundus image restoration on a new clinical benchmark. JBHI 2022
>
> [78] Video-rate hyperspectral camera based on a CMOS-compatible random array of Fabry–Pérot filters. Nature Photonics 2023

---

> > ### Comment · Reviewer_kC2s · 2023-08-17
> > **Thanks for the response**
> >
> > The response from the authors has well addressed my concerns. I also read the comments from other reviewers. Now I am convinced that the work has both novelty and contributions to the fields of both compact NN design and spectral compressive imaging. Therefore, I would like to raise my score to Accept.

---

### Official Review · Reviewer_ZUfp · 2023-06-28

**Soundness:** 4 excellent
**Presentation:** 4 excellent
**Contribution:** 4 excellent
**Rating:** 7
**Confidence:** 5

**Summary:**

This paper proposes a novel method, Binarized Spectral-Redistribution Network (BiSRNet), for efficient and practical HSI restoration from compressed measurement in snapshot compressive imaging (SCI) systems. This paper redesigns the base model and presents the basic unit (BiSR-Conv) for model binarization. Specifically, this convolutional layer of BiSRNet is tailored for hyperspectral image processing. Comprehensive quantitative and qualitative experiments show that the proposed BiSR-Net outperforms state-of-the-art binarization methods and achieves comparable performance with full-precision algorithms.

**Strengths:**

(1) This paper redesigns a U-Net consisting of the four BCMs as shown in Figure 4 to let the full-precision information flow pass by the all network. This point is critical since previous BNNs do not consider all the situations in feature reshaping. Then in the basic unit, BiSR-Conv, with the insight of treating different HSIs with different densities and distribution, this paper proposes to shift HSIs before binarization to allow more binarized HSI activation.

(2) Comprehensive experiments have been conducted including synthetic and real experiments to demonstrate the superiority of BiSRNet. The ablation study is also extensive. The performance of BiSRNet is impressive, which surpasses existing SOTA BNNs by huge margins, over 2.5 dB.

**Weaknesses:**

(1) Some details are missing. For example, how did you get the 3D mask $\mathbf{M}^*$? I remember the coded aperture used in CASSI is 2D. Its shape should be $H \times W$. Why does it have an additional dimension here? And how?

(2) Some critical experimental analyses are lacking. For instance, at the end of section 4.4, “Binarizing Different Parts”, why binarizing the bottleneck can reduce the most parameters? And why binarizing the decoder can achieve the largest Ops reduction? An analysis should be provided to explain this.

(3) I remember Binary Connect [39] binarizes the weights of CNN. But it seems that the activations of Binary Connect [39] are also binarized in Table 1. Why did you do that? Could you please explain this?


**Questions:**

Please refer to Weaknesses

**Limitations:**

This paper proposes BiSRNet for efficient and practical HSI restoration from compressed measurement in snapshot compressive imaging (SCI) systems. But this paper has not discussed the limitations of the proposed method. I suggest the author discuss the potential and limitations of the proposed method in more detail, which will make the contributions of the proposed method more significant.

---

> ### Author Rebuttal · Authors · 2023-08-04
>
> &nbsp;
>
> ### Response to Reviewer ZUfp
>
> &nbsp;
>
> Thanks for your valuable comments.
>
> Code and models will be released to the public.
>
> &nbsp;
>
> `Q-1:` How to derive the 3D mask? Why does it have an additional dimension?
>
> `A-1:` The original coded aperture $\mathbf{M}^* \in \mathbb{R}^{H \times W}$ is a 2D mask. However, as analyzed in Sec. 1 (Line 9 - 33) of the supplementary, the 3D HSI cube is point-wisely modulated by $\mathbf{M}^*$ in each spectral channel before dispersion. To simply describe the process of modulation and dispersion, we derive a 3D Mask $\mathbf{M} \in \mathbb{R}^{H \times (W + d(N_{\lambda}-1)) \times N_{\lambda}}$ by shifting  $\mathbf{M}^*$ as
>
> $\mathbf{M} (u,v, n_{\lambda}) = \mathbf{M}^* (x, y + d(\lambda_n - \lambda_c))$
>
>
> Similarly, we obtain $\tilde{\mathbf{F}} \in {\mathbb R}^{H \times (W + d(N_{\lambda}-1)) \times N_{\lambda}}$ by shifting the original HSI signal $\mathbf{F}$ as
>
> $\tilde{\mathbf{F}} (u,v, n_{\lambda}) = \mathbf{F} (x, y + d(\lambda_n - \lambda_c), n_{\lambda})$
>
>
> Following this, we can simply reformulate the measurement $\mathbf{Y}$ as
>
> $\mathbf{Y} = \sum\nolimits_{n_{\lambda}=1}^{N_{\lambda}}   \tilde{\mathbf{F}} (:,:, n_{\lambda})  \odot \mathbf{M} (:,:, n_{\lambda}) + \mathbf{E}$.
>
>
> &nbsp;
>
>
> `Q-2:` Why binarizing the bottleneck can reduce the most parameters? And why binarizing the decoder can achieve the largest Ops reduction? An analysis should be provided to explain this.
>
> `A-2:` Thanks for reminding. We will add the following explanation in the revision.
>
> (1) As shown in Fig. 2 (a) of the main paper, when the feature map is downscaled in the encoder, its channel number is doubled. Thus, the feature map in the bottleneck has the most channel number, leading to the most parameters in the bottleneck part of the network.
>
> (2) As shown in Fig. 4 (a) and (b) of the main paper, the convolution of downsample modules in encoder follows the downscaled operation while the convolution of upsample modules in decoder follows the upscaled operation. This implies the input spatial size of the convolution in upsample modules at the same level is larger than that of the convolution in downsample modules, which requires more computational costs. Thus, binarizing decoder achieves the largest Ops reduction.
>
> &nbsp;
>
> `Q-3:` Why binarizing the activations of Binary Connect in Tab. 1?
>
> `A-3:` The activations and weights of all compared BNN methods in Tab. 1 of the main paper are binarized for fair comparison. Thus, we also binarize the activations of Binary Connect.
>
> &nbsp;
>
> `Q-4:` Limitations of our work
>
> `A-4:` In fact, we have discussed the limitations in Sec. 3 (Line 59 - 64) of the supplementary. The main limitation of our work is that the model binarization sacrifices the HSI reconstruction performance. More specifically, compared to the full-precision counterpart, our BiSRNet is 4.35 (34.11 - 29.76) dB lower in PSNR and 0.099 (0.936 - 0.837) lower in SSIM. The PSNR and SSIM are reduced by 12.8\% and 10.6\%, respectively. However, this performance drop is smaller than that of other model binarization methods. To handle this issue, we will study how to preserve more performance while reducing the memory and computational complexity as much as possible in model binarization.

---

> > ### Comment · Reviewer_ZUfp · 2023-08-16
> > **Thank you for addressing my comments**
> >
> > Thank you for addressing my comments, especially on 3D mask derivation. The paper offers many insights into binarized spectral compressive imaging. I'll keep my score of 7. Thank you.

---

### Official Review · Reviewer_nobf · 2023-06-29

**Soundness:** 4 excellent
**Presentation:** 4 excellent
**Contribution:** 4 excellent
**Rating:** 7
**Confidence:** 4

**Summary:**

This work focuses on studying the binarized spectral compressive imaging reconstruction problem. A Binarized Spectral-Redistribution Network (BiSRNet) is proposed. The authors first design a basic U-Net as the base model to begin the binarization. Then a Binarized Spectral-Redistribution Convolutional (BiSR-Conv) unit is proposed to replace the 32-bit convolutional layer of the base model to derive BiSRNet. The BiSR-Conv has two advantages than vanilla 1-bit convolutional layer: (i) can redistribute the spectral distributions before binarization. (ii) has a scalable hyperbolic tangent function to approach the Sign function more closely.

**Strengths:**

 The novelty and motivation are good. First of all, the research topic is new. Nobody explores binarized spectral compressive imaging before. The authors not only contribute a good method but also conduct experiments using before BNNs designed for other topics. We should respect this. Secondly, the redesigned base model is well motivated. The architecture shows the insight of “full-precision information flow”. The idea of redistributing HSI representations before binarization is interesting and reasonable. The Tanh(\alpha x) is also amazing. Flexibly controlling the gap with Sign function is really cool.

+ The performance is good and solid. The proposed BiSRNet outperforms the SOTA BNNs widely used in image classification. Meanwhile, the computational and memory complexity of BiSRNet is much lower than full-precision CNNs while the performance is comparable. In the real HSI experiments, it seems that the proposed BiSCI performs better than CNN-based methods in noise suppression.

+ The experiments are comprehensive, not only quantitative but also qualitative comparisons shows the advantages of the proposed method. The ablation study is sufficient to verify the effectiveness of the proposed techniques.

+ The codes and pre-trained weights are provided in the supplementary. The reproducibility can be checked and ensured.

+ The writing is good and easy-to-follow. The figures, tables, and mathematical notations are very clear.

**Weaknesses:**

- The definition is a little confusing. Sign(0) = 0 and Tanh(0) = 0. But in this paper, Sign(0) and Tanh(0) is defined as -1, as shown in Eq.(3) and line 152. Why? More explanation should be provided.

- Although the proposed BNN achieves good performance with CNN. But compared with Transformer-based methods, there is a large gap. I understand that binarizing Transformer is not easy because of the self-attention mechanism. But it is good idea to keep trying.


**Questions:**

Will you also plan to release the code of other BNNs compared in Table 1?

**Limitations:**

See the above comments

---

> ### Author Rebuttal · Authors · 2023-08-04
>
> &nbsp;
>
> ### Response to Reviewer nobf
>
> &nbsp;
>
> Thanks for your valuable comments.
>
> Code and models will be release to the public.
>
> &nbsp;
>
> `Q-1:` Why Sign(0) and Tanh(0) are defined as -1?
>
> `A-1:` As explained in Line 150 -152 of the main paper, if strictly following the mathematical definition, Sign(0) $= 0 \neq \pm 1$. However, in BNN, the weights and activations are binarized into 1-bit, i.e., only two values ($\pm 1$). Hence, Sign(0) is usually set to $\pm 1$, like [33, 34, 35, 36, 37, 38, 52]. Following this common setting, we also define $\underset{\alpha \rightarrow +\infty}{lim}~\text{Tanh}(\alpha \cdot 0) = -1$ in BNN.
>
> &nbsp;
>
> `Q-2:` It is a good idea to keep trying on binarizing Transformer.
>
> `A-2:` Thanks for your reminding. As you mentioned, Transformer is tough to binarize because of the computation scheme of self-attention. According to the experience, preserving at least 8-bit weights and activations can considerably control the performance gap between quantized and full-precision (32-bit) Transformers. We would like to set your advice as a future research direction and have a try.
>
> &nbsp;
>
> `Q-3:` Will we plan to release the code of other BNNs compared in Table 1?
>
> `A-3:` Yes, of course. We will release all of them. Our goal is to establish a toolbox and baseline for further research in this topic.

---

> > ### Comment · Reviewer_nobf · 2023-08-16
> >
> > After reading the rebuttal, my concerns have been well solved. Considering the novelty and solid experiments, I tend to keep my original score as "Accept".

---

### Official Review · Reviewer_aGHs · 2023-07-05

**Soundness:** 3 good
**Presentation:** 4 excellent
**Contribution:** 4 excellent
**Rating:** 6
**Confidence:** 4

**Summary:**

In this paper, binarized neural network is first utilized in hyperspectral image reconstruction. For model binarization, authors propose the Binarized Spectral-Redistribution Convolution (BiSR-Conv), which adaptively redistributes the HSI representations before binarizing activation. Since the Sign function is non-differentiable, a scalable hyperbolic tangent function is applied in backpropagation, which has less approximation error. In BiSR-Conv, the additional residual connection encourages the full-precision information to propagate throughout the whole network. Based on BiSR-Conv, four binarized convolutional modules are designed to address the dimension mismatch issue during feature reshaping.

**Strengths:**

An application to a new domain;
Clear and consecutive writing;
A feasible idea for deploying spectral compression imaging in edge devices.


**Weaknesses:**

Experiments are insufficient. How does the ‘full-precision information’ affect the whole network? Is the ‘full-precision information’ play a dominant role in BiSR-Conv or not?

**Questions:**

In Sign Approximation experiment, the proposed scalable hyperbolic tangent function and previous Sign approximation functions should be compared in other BNNs, to further verify the generalization of the proposed function.

**Limitations:**

yes

---

> ### Author Rebuttal · Authors · 2023-08-04
>
> &nbsp;
>
> ### Response to Reviewer aGHs
>
> &nbsp;
>
> Thanks for your valuable comments.
>
> Code and models will be released to the public.
>
> &nbsp;
>
> `Q-1:` How does the 'full-precision information' affect the whole network? Does the 'full-precision information' play a dominant role in BiSR-Conv or not?
>
> `A-1:` In BNNs, the weights and activations are binarized into 1 bit. Thus, there is a large gap between the outputs of the binarized and full-precision convolution, which severely degrades the HSI reconstruction performance. Hence, it is very important to allow the full-precision information flow to compensate for this quantization error.
>
> (1) To this end, we do not binarize the first (embedding) and last (mapping) convolution modules of the base model in Fig. 2 (a) of the main paper. Then the full-precision representations can be input into the binarized encoder and full-precision derivatives can be backpropagated to the binarized decoder.
>
> (2) In addition, to allow full-precision information flow through all binarized convolution layers, we design BiSR-Conv unit and the four binarized convolution modules, as shown in Fig. 4 of the main paper. More specifically, we add a bypass identity path in BiSR-Conv unit to propagate full-precision information, as shown in Fig. 2 (c). Besides, in Sec. 3.3, our binarized convolution modules cleverly use channel-wise concatenating and splitting operations to free the intermediate feature maps at the input and output ends of BiSR-Conv from being reshaped. In this case, the full-precision information flow is not blocked as the normal convolution modules do, as shown in Fig. 4.
>
> (3) The experiments to verify the importance of the full-precision flow are reported in Tab. 2 (a) of the main paper. Baseline-1 adopts vanilla 1-bit convolution and the normal convolution modules in Fig. 4. The full-precision information in the binarized parts is impeded. As a result, Baseline-1 only achieves poor results, 23.90 dB. When we successively apply the proposed BiSR-Conv and the four binarized modules, the full-precision information can be propagated through all layers of the BNN, leading to significant improvements of 3.90 and 1.96 dB.
>
> (4) Besides, we also conduct more experiments to verify the importance of full-precision information in the following table. When we binarize the feature embedding and mapping blocks in Fig. 2 (a), BiSRNet degrades by 2.13 and 1.06 dB respectively. When we jointly binarize the two blocks, the full-precision information can not be forward and backward propagated. As a result, the reconstruction performance degrades by 2.87 dB.
>
> The above results and analysis reveal the significant importance of full-precision information. This is also the key insight and motivation to design our BiSRNet and the four binarized convolution modules in Fig. 4.
>
> &nbsp;
>
> `Q-2:` In Sign approximation experiment, the proposed scalable hyperbolic tangent function and previous Sign approximation functions should be compared in other BNNs, to further verify the generalization of the proposed function.
>
> `A-2:` Thanks for your suggestion. Following your advice, we conduct experiments of Sign approximation in other BNNs. The PSNR (dB) results are reported in the following table.
>
> | Methods | BiConnect | BNN | Bi-Real | IRNet | ReAcNet | BBCU | BTM | BiSRNet | Avg |
> | :-: | :-: | :-: | :-: | :-: | :-: | :-: | :-: | :-: | :-: |
> | Clip($x$) | 22.19 | 23.88 | 26.20 | 26.16 | 26.41 | 26.43 | 27.15 | 28.97 | 25.92 |
> | Quad($x$) | 22.34 | 23.95 | 26.26 | 26.30 | 26.48 | 26.51 | 27.21 | 29.02 | 26.01 |
> | Tanh($\alpha x$) | 22.93 | 24.46 | 26.94 | 26.95 | 27.23 | 27.25 | 27.97 | 29.76 | 26.69 |
>
> Our scalable hyperbolic tangent function achieves 0.77 and 0.68 dB improvements on average than the piecewise linear and quadratic functions, showing the effectiveness of our proposed technique.

---

> > ### Comment · Area_Chair_PQMi · 2023-08-17
> >
> > Dear Reviewer,
> > Please take a look at the response from authors to your comments made in your review and update your final score.
> > Thanks,
> > AC

---

### Official Review · Reviewer_F7DC · 2023-07-06

**Soundness:** 3 good
**Presentation:** 3 good
**Contribution:** 3 good
**Rating:** 6
**Confidence:** 3

**Summary:**

This paper proposes a Binarized Neural Network based approach known as BiSRNet for binarized HSI restoration. The main motivation of the paper stems from the fact that any CNN or transformer-based architectures are computationally heavy for devices with low computing power and hence need extremely fast and light weight networks like binary neural networks. This paper redesigns the binary conv layer (BiSR-Conv) to exploit the distributed nature of HSI representations. They also employed a scalable Tanh function to decrease the approximation error. Overall, this paper is well-written and reasons well for various changes to the design of BCNNs.

**Strengths:**

- Well-motivated problem and design of the solution
- Good empirical evidence
- Good presentation
- Clear to understand

**Weaknesses:**

- Main weakness of this paper is their redesigned conv modules are only applicable to the HSI domain? If so this limits the scope and impact of this work.
- Lack of theory


**Questions:**

- I am not very familiar with HSI work so I do not have many questions at this point.

**Limitations:**

Please see the weakness.

---

> ### Author Rebuttal · Authors · 2023-08-05
>
> &nbsp;
>
> ### Response to Reviewer F7DC
>
> &nbsp;
>
> Thanks for your valuable comments.
>
> Code and models will be released to the public.
>
> &nbsp;
>
> `Q-1:` Are the redesigned conv modules only applicable to the HSI domain?
>
> `A-1:` No. Although this work mainly studies the binarized spectral compressive imaging reconstruction problem in HSI domain, the proposed method can be generalized to other natural and even medical image domains.
>
> (1)	First of all, in Line 131 – 138 of the main paper, we notice that HSI signals have different density and distribution along the spectral dimension due to the constraints of specific wavelengths. To adaptively fit this HSI nature, we propose to redistribute the HSI representations along the spectral dimension before binarizing the activation. When this technique is used to other image domains, it shifts and reshapes the feature maps in channel wise to preserve more information in activation binarization.
>
> (2)	Secondly, in Line 138 – 160 and Fig. 3 of the main paper, our designed scalable hyperbolic tangent function is not only applicable to the HSI domain. Instead, it is a very basic and general technique to approximate the Sign function in any BNN for any image domain.
>
> (3)	Thirdly, allowing full-precision information flow is a critical insight for us to design BiSR-Conv unit in Fig. 2 (c) and the four binarized convolution modules in Fig. 4. The identity paths propagate the full-precision information to all layers of the network. By this means, the quantization error between BNN and CNN can be narrowed down.
>
> The proposed BiSR-Conv and the four binarized convolution modules in Fig.4 can replace the vanilla binarized convolution layer and normal downsample / upsample / fusion convolution modules (in Fig. 4) in other BNNs and can be seamlessly generalized to other image domains.
>
> We conduct experiments of RGB image denoising ($\sigma = 25$). The BNNs are trained on DIV2K [73] and tested on CBSD68 [74], Kodak24 [75], and Urban100 [76]. Besides, we also conduct experiments of medical image enhancement on Real Fundus [77] dataset. The PSNR (dB) results are shown in the following table.
>
> | Datasets| BNN | Bi-Real | IRNet | BTM | ReAcNet | BBCU | BiSRNet |
> | :- | :-: | :-: | :-: | :-: | :-: | :-: | :-: |
> | CBSD68 | 22.67 | 28.72 | 29.01 | 29.91 | 29.95 | 30.56 | **31.15** |
> | Kodak24 | 22.58 | 29.17 | 29.54 | 30.64 | 30.65 | 31.28 | **32.06** |
> | Urban100 | 22.67 | 28.18 | 28.35 | 29.05 | 29.20 | 29.96 | **30.21** |
> | Real Fundus | 16.89 | 23.94 | 24.03 | 25.58 | 24.16 | 24.25 | **26.31** |
>
> Our method still significantly outperforms other BNNs. These results demonstrate the generality and effectiveness of our method on RGB/Medical image domains. The reason why we focus on studying binarized spectral compressive imaging reconstruction is that this problem has not been studied until now. We aim to fill this research gap. All compared methods in Tab. 1 are re-implemented by us.
>
> &nbsp;
>
> `Q-2:` Lack of theory
>
> `A-2:` This work mainly studies an application problem, i.e., binarized spectral compressive imaging reconstruction, instead of theory. The first keyword labeled on this paper is “Applications” and the primary area of this submission is “Machine Vision” instead of “Machine Learning Theory”.
>
> However, there are still some theoretical analysis and derivation in our paper.
> In Line 138 – 160 of the main paper, we first theoretically prove that the proposed scalable hyperbolic tangent function can arbitrarily approach the Sign function in Eq. (6) and (7). Then we theoretically compute and compare the approximation error of previous picewise linear and quadratic functions and our scalable hyperbolic tangent function in Line 143 – 145, Fig. 3, and Eq. (8). Finally, we theoretically analyze the differentiability and flexibility of previous and our approximation functions to show the advantages of our method.
>
> Besides, we also provide detailed theoretical derivation of the CASSI mathematical model in Sec. 1 (Line 8 – 48) of the supplementary.
>
> &nbsp;
>
> **References**
>
> [73] NTIRE 2017 Challenge on Single Image Super-Resolution: Dataset and Study. CVPRW 2017.
>
> [74] A database of human segmented natural images and its application to evaluating segmentation algorithms and measuring ecological statistics. ICCV 2001.
>
> [75] Residual learning of deep convolutional neural networks for image denoising. Journal of Intelligent & Fuzzy Systems, 2019.
>
> [76] Single image super-resolution from transformed self-exemplars. CVPR 2015.
>
> [77] Rformer: Transformer-based generative adversarial network for real fundus image restoration on a new clinical benchmark. JBHI 2022.

---

> > ### Comment · Area_Chair_PQMi · 2023-08-17
> >
> > Dear Reviewer,
> > Please take a look at the response from authors to your comments made in your review and update your final score.
> > Thanks,
> > AC

---

### Decision · Program_Chairs · 2023-09-21

**Decision:**

Accept (poster)

**Comment:**

All reviewers found some merit in this work.